# “My Friends Would Believe My Word”: Appropriateness and Acceptability of Respondent-Driven Sampling in Recruiting Young Tertiary Student Men Who Have Sex with Men for HIV/STI Research in Nairobi, Kenya

**DOI:** 10.3390/ijerph19127331

**Published:** 2022-06-15

**Authors:** Samuel Waweru Mwaniki, Peter Mwenda Kaberia, Peter Mwangi Mugo, Thesla Palanee-Phillips

**Affiliations:** 1School of Clinical Medicine, Faculty of Health Sciences, University of the Witwatersrand, Johannesburg 2000, South Africa; 2University Health Services, Central Administration, University of Nairobi, Nairobi P.O. Box 30197-00100, Kenya; 3School of Mathematics, Faculty of Science and Technology, University of Nairobi, Nairobi P.O. Box 30197-00100, Kenya; petermwenda592@gmail.com; 4Wellcome Trust Research Programme—Kenya Medical Research Institute, Nairobi P.O. Box 54840-00200, Kenya; pmugo@kemri-wellcome.org; 5Wits Reproductive Health and HIV Institute, Faculty of Health Sciences, University of the Witwatersrand, Johannesburg 2000, South Africa; tpalanee@wrhi.ac.za; 6Department of Epidemiology, School of Public Health, University of Washington, Seattle, WA 98195, USA

**Keywords:** human immune deficiency virus, sexually transmitted infections, men who have sex with men, respondent-driven sampling, sub-Saharan Africa, Kenya

## Abstract

(1) Background: We conducted formative research to assess the appropriateness and acceptability of respondent-driven sampling (RDS) in recruiting tertiary student men who have sex with men (TSMSM) into a prospective human immunodeficiency virus/sexually transmitted infection (HIV/STI) biobehavioral survey in Nairobi, Kenya. (2) Methods: Between September and October 2020, semi-structured qualitative interviews were held with service providers from organizations that serve MSM (n = 3), and TSMSM (n = 13). Interviews were conducted in English, audio-recorded and transcribed, then thematically analyzed using NVivo version 11. (3) Results: Service providers reflected that RDS was appropriate due to the large though concealed networks of TSMSM. TSMSM perceived RDS to be acceptable based on their large social network sizes and the trust that existed amongst themselves. TSMSM were concerned about participating due to the risk of being outed as MSM and hence emphasized that researchers needed to assure them of their confidentiality and include MSM as part of the study team to encourage participation. (4) Conclusions: RDS was perceived as both an appropriate and acceptable sampling method. Use of RDS should be considered as a strategy for recruiting young, marginalized populations for HIV/STI research.

## 1. Introduction

Globally, the HIV epidemic remains poorly defined among young MSM (YMSM) aged 15–24 years. There is minimal data on population size estimates of YMSM and their HIV rates, as well as risk and protective factors. This is partially attributed to a lack of research and surveillance, as well as the difficulty of reaching YMSM with services [1], especially in countries such as Kenya, where same-sex behavior continues to be criminalized [2]. Additionally, YMSM are vulnerable to HIV due to widespread discrimination and stigmatization, homophobia and violence, power imbalances in relationships and alienation from family and friends [3]. Largely due to these factors, YMSM are invariably at disproportionate risk of acquiring HIV/STI compared to young heterosexual males or older MSM [4].

During the transition period from adolescence to adulthood and a parallel shift from secondary to tertiary educational institutions, a majority of students find themselves in an environment with limited direct supervision of their behavior and increased freedom and opportunities to engage in risky behavior, such as alcohol use and casual sex [5]. For tertiary student MSM (TSMSM), this involves a sudden exposure to peers with similar sexual orientation, increased freedom and autonomy in young adulthood and more socializing opportunities [6]. Subsequently, they may engage in sex more frequently [7], and some might have unprotected anal intercourse more regularly [8], increasing their risk of HIV/STI infection. With most tertiary students in Kenya owning smart phones [9], TSMSM, like other MSM, may also use geosocial applications, such as Grindr^®^, to seek more sex partners for immediate sexual gratification without paying attention to the accompanying risk of HIV/STI infection [10].

Even in countries where same-sex behavior is legal and data are available, TSMSM have been found to engage in high-risk behavior that increases their susceptibility to HIV/STI. A systematic review and meta-analysis of studies published between 2003 and 2016 in China found out that TSMSM engaged in sex work, group sex, sex with female partners, drug use and seeking casual sex partners online [11]. In the USA, TSMSM have been shown to have complex and expansive sexual networks within which HIV/STI may circulate widely and magnify epidemics [12]. In South Africa, a behavioral survey among TSMSM showed a high prevalence of various risk factors for HIV transmission among TSMSM, including high partner turnover, concurrent sexual partners, presence of STI, early sexual debut, having female sex partners, forced sex experiences, inconsistent condom usage, and alcohol and drug use [13].

In Kenya, same-sex behavior is criminalized [2]. This may further amplify the impact of behavioral risk taking on HIV/STI transmission over and above what has been observed among TSMSM in countries where same-sex behavior is legal [11,12,13]. A literature review of TSMSM in Kenya yielded no information on the prevalence of HIV/STI or risk behaviors in this potentially at-risk sub-group of MSM. There is a need to conduct research to better understand the epidemic in this neglected sub-population, informing the form and design of interventions to combat the HIV/STI syndemic in this population.

Respondent-driven sampling (RDS) has become a popular method to assess HIV/STI and risk behaviors in hidden or hard-to-reach populations, such as MSM. RDS is both a recruitment and statistical method that utilizes “snowball sampling”, where individuals recruit those they know; the recruited individuals in turn recruit those they know and so on until an adequate sample is reached. This recruitment is combined with a mathematical model that weights the sample to compensate for the non-random sampling [14]. Using RDS makes it possible to draw statistically valid samples from populations for whom there is no sampling frame and provide unbiased population estimates and confidence intervals of disease prevalence and risk behaviors, such as unprotected anal intercourse, among MSM [15]. Recruitment begins with ‘seeds’ purposely selected from the target population and given, at most, three ‘coupons’ to recruit their peers into the study. Participants who are recruited by the ‘seeds’ form the first ‘wave’ of recruitment. Participants in the first ‘wave’ are then issued with ‘coupons’ to recruit the second ‘wave’ of participants. This process goes on until the calculated sample size is reached. To encourage recruitment, participants are given an incentive to complete the survey (primary incentive) and to recruit their peers (secondary incentive) [16].

Previous studies that investigated HIV/STI in TSMSM recruited participants through either convenience sampling [17,18,19] or conducted studies among the general MSM population, then drew a subset of TSMSM from this sample [20] or conducted studies among the general tertiary student population and drew a subset of TSMSM from this sample [7,13,21].

Formative research helps researchers and public health practitioners reach their target population and define that population’s characteristics that are pertinent to a particular public health issue of interest [22]. Assessing appropriateness and acceptability is important so as to gauge the suitability of a particular method for a given population and the agreeability of that method to a range of stakeholders, respectively [23]. We conducted formative research to investigate these two outcomes for RDS as a recruitment strategy for TSMSM in a proposed HIV/STI biobehavioral survey [24], as well as to identify potential challenges that would be faced with the use of such a method and proposed strategies to mitigate these challenges. The study was conducted in Nairobi, the capital of Kenya. Our study showed that RDS is both appropriate and acceptable and should be considered as a strategy for recruiting young and marginalized populations for HIV/STI research.

## 2. Materials and Methods

### 2.1. Aim

The primary aim of the study was to assess the appropriateness and acceptability of using the RDS method to recruit TSMSM into a prospective HIV/STI biobehavioral survey. The secondary aim of our study was to identify potential challenges that would likely be encountered with the use of the RDS method, as well as proposed mitigation strategies to address these challenges.

### 2.2. Theoretical Framework and Reporting

The study was underpinned by the content analysis theoretical framework, where data is systematically organized into a structured format [25]. Reporting of the study was done following the consolidated criteria for reporting qualitative studies (COREQ) [26]. (See Appendix A).

### 2.3. Site

The study was conducted in Nairobi, the capital city of Kenya and home to more than 50 campuses of various universities and technical and vocational education training (TVET) institutions.

### 2.4. Participants

Service providers were purposely selected from community-based organizations (CBOs) that provide MSM-exclusive health services (n = 2) and from a non-governmental organization (NGO) that provides MSM-friendly health services (n = 1). The three organizations were selected purposely, first, because they are the main providers that offer MSM-exclusive/friendly health services in Nairobi and are popular with both YMSM and TSMSM. Secondly, we sought to obtain information on how to best reach different typologies of TSMSM, with the ‘more out’ TSMSM assumed to be more likely to use the MSM-exclusive services and those ‘less out’ to use the non-exclusive but MSM-friendly services. Thirdly, the three organizations are conveniently located, with the NGO within and the CBOs located about 5 km from the central business district, hence easily accessible by TSMSM.

TSMSM were purposely recruited with the help of the service providers at the three facilities. After completing the interviews, the service providers were given the telephone contact of one member of the study team and asked to request TSMSM known to them to contact the member of the study team regarding possible participation in the formative research. The service providers were given the ideal qualities of the TSMSM the study team was interested in speaking to, which included peer leaders with a social network size of at least nine TSMSM interested in research and working with the TSMSM community. Consequently, contact was established with 16 TSMSM, out of which 13 were interviewed. Of the three who were not interviewed, one had already completed university education, one was not studying in Nairobi and the third was unable to create time for the interview.

### 2.5. Data Collection

Semi-structured interview guides were obtained from the Integrated HIV Bio-behavioral Surveillance Toolbox of the University of California, San Francisco [27] and modified to fit the purpose of the current study (see Appendix A). All the interviews were conducted by the first author, who introduced himself to the participants and provided them with detailed information about the study, including the key components of the RDS method. Interviews with service providers consisted of five major domains, namely service providers’ roles in their respective organization, type of services offered by the organizations, description of the TSMSM community and their networks, appropriateness and acceptability of using RDS to recruit participants for the HIV/STI biobehavioral survey, challenges expected with the use of RDS and proposed strategies to mitigate these challenges. Interviews with TSMSM consisted of three major domains, namely their social networks, appropriateness and acceptability of using RDS to recruit participants for the planned HIV/STI biobehavioral survey, challenges expected with the use of RDS and proposed strategies to mitigate these challenges. The network size was estimated by asking the question: ‘How many TSMSM do you know by name, and they know you by name, and they live in and around the city of Nairobi?’. Demographic information was also collected using a brief self-administered questionnaire on REDCap (Vanderbilt University, Nashville, TN, USA).

Interviews with the service providers were held at their respective workplaces. Interviews with the TSMSM were held in a private room at the integrated counselling and education center of the University of Nairobi health services department. Only the interviewer and one participant were present at the time of each interview. Interviews with the service providers lasted an average of 40 min, whereas those with TSMSM took an average of 35 min. For the interviews with TSMSM, saturation was reached by the 10th participant. All interviews were held between September and October 2020.

The interviewer took notes during all the interviews. All interviews were audio-recorded, transcribed and translated to English in the few instances where Swahili was used. The interviewer then read the transcripts as he listened to the audio recordings to ensure that the transcripts and translations, if any, were accurate.

### 2.6. Data Analysis

Data in the form of the transcripts was managed using NVivo software version 11 (QSR International). A thematic framework approach was adopted. Analysis was conducted both deductively and inductively, using themes set a priori and coding emerging themes, respectively. Themes were supported with verbatim excerpts from the transcripts. Coding was performed independently by two members of the study team, who then compared codes for agreement and decided whether to merge some codes, get rid of others or come up with new ones.

### 2.7. Trustworthiness of the Data

The trustworthiness of the data was premised on the following criteria: credibility, transferability, dependability and confirmability [28,29]. To ensure credibility, the study used well-established methods for qualitative investigation. Prior to data collection, the first author familiarized himself with the setup and roles of the participating CBOs/NGO by reading content posted on the organizations’ respective websites. During data collection, iterative questioning was deployed by rephrasing previously asked questions to gain deeper understanding of and confirm the information provided by participants at the first instance of questioning. Member checking was performed “on the spot” by paraphrasing and summarizing what the participants said at intervals and at the end of the interviews, respectively. For transferability, details about the study participants, study design, data collection procedures and analysis have been made available to aid in comprehension of the findings and comparison/contrasting of these findings with those of similar studies. To address dependability, the study processes are reported in sufficient detail to enable future researchers reproduce the study, even if not necessarily to obtain similar results. For confirmability, individual interviews were used due to the sensitivity of the study topic occasioned by criminalization and marginalization of same-sex behavior in Kenya. Confirmability was also established by supporting the findings with verbatim quotes from the participants.

### 2.8. Ethics Statement

The study protocol was reviewed and approved by the University of the Witwatersrand Human Research Ethics Committee-Medical (reference number: M200215) and the University of Nairobi-Kenyatta National Hospital Ethics and Research Committee (reference number: P990/12/2019). The study also received a letter of support from the Key Populations Program of the National AIDS and STI Control Program (NASCOP), Ministry of Health, Government of Kenya. Participants provided written informed consent before taking part in the study.

## 3. Results

### 3.1. Participant Characteristics

The service providers who participated from the CBOs were both male, in their 30s and identified as gay. The service provider from the NGO was female, in her 40s and identified as heterosexual. All the providers had college degrees. All the service providers had dual roles at their respective organizations. The service provider at the first CBO worked as a pre-exposure prophylaxis (PrEP) program officer/research coordinator, the one at the second CBO as a community mapping officer/peer educator supervisor and the one at the NGO as a clinician/head of the clinic. These service providers were selected on the basis of their day-to-day interaction with members of the MSM population and the assumption that they would have rich information on the TSMSM sub-population who were part of their clientele. All three organizations offered HIV/STI prevention, treatment and care services. Additionally, the two CBOs offered other services, including community outreach, as well as economic empowerment of and provision of legal aid to the MSM communities they served. On the other hand, the NGO offered care to victims of gender-based violence and sexual and reproductive health services to other key populations, such as female sex workers. The service providers referred 16 TSMSM to the research team, out of which 13 participated in the formative research (n = 13).

Demographic characteristics of the TSMSM participants are presented in Table 1. More than three-quarters (76.9%) were 21–24 years of age. All participants were undergraduate students in their first (23.1%), second (38.5%) or third (38.5%) year of study. A majority (69.2%) identified as homosexual or gay and the rest as bisexual.

The following sections explore the main themes from the analysis of the interviews with both the service providers and TSMSM. The themes include (a) social networks of TSMSM as a measure of the appropriateness of using the RDS method for recruiting TSMSM into the prospective HIV/STI biobehavioral survey; (b) appropriateness and acceptability of using the RDS method as perceived by participants; and (c) likely challenges anticipated with the use of the RDS method, as well as proposed strategies to mitigate these challenges. Themes are described and supported with illustrative quotes from the interviews. Quotes are attributed to participants using pseudonyms and role (service provider or student).

### 3.2. Social Networks of TSMSM

A key theme that emerged from the interviews is that TSMSM constituted a large and well-networked sub-group of the MSM community. TSMSM formed networks among themselves, as well as with other sub-groups of MSM. TSMSM had regular face-to-face interactions with their peers from the same institution and those from different institutions. However, since the outbreak of COVID-19, these interactions had been hampered by government-imposed restrictions to reduce risk of disease transmission. Consequently, TSMSM mainly interacted through social media applications, such as Facebook, WhatsApp, Twitter and Instagram.

“In the student MSM community, what happens is that there is some sort of invisible network that is hidden but it is very large and well connected.”Justin, service provider.

“I know about 30 to 40 people (other TSMSM in Nairobi). I have seen about 10 (in the last 30 days)… due to the COVID situation, so most people travelled (out of Nairobi)… most of them are the people we study with, like in the same institution… but I also know others from other universities.” Kyle, student.

“Twenty-five to thirty (number of TSMSM known). Due to the corona (COVID-19), I have seen about 12… we are used to each other… we always chat through WhatsApp, Facebook and Twitter sometimes… even through Instagram.”Caleb, student.

TSMSM exhibited strong ties with their peers (intragroup ties), through which they found safe spaces to socialize, such as night clubs and house parties. The house parties offered opportunities to engage in risky behaviors, such as use of alcohol and casual sex. In spaces where there were other MSM, such as night clubs, TSMSM tended to interact and socialize with each other, staying away from other MSM perceived to have lower educational attainments.

“They (TSMSM) love parties… like on Fridays, they have house parties among themselves… so they really interact among themselves so much, then they get drunk and have sex.”Nicole, service provider.

“Most of them (MSM) group themselves into clusters, whereby even if you get into a club, you will find them like the tertiary ones (students) are seated at one corner, and the lower end ones (with lower education levels) at the other corner.”Mike, service provider.

TSMSM also interacted with other sub-groups of MSM. TSMSM were a bridging sub-population between well-off, older MSM and less well-off MSM within their age group and engaged in transactional sex, receiving money from the former and in turn giving money to the later.

“MSM students have sponsors (older and well-off MSM)… because they want to have a flashy lifestyle, they want to have money, so they will have sponsors to sponsor their lifestyles.”Nicole, service provider.

“Tertiary MSMs, they have sex with those guys with money so that they can cater for their needs. Once (after) they are from that upper end, they will come into the clubs… they try to mingle and buy sex from the lower end partners.”Mike, service provider.

### 3.3. Appropriateness and Acceptability of Using RDS to Recruit TSMSM in a Prospective HIV/STI Survey

Participants felt that RDS would be an appropriate and acceptable method of recruiting TSMSM for the prospective HIV/STI survey. TSMSM regarded RDS as an appropriate recruitment strategy due to the hidden but networked nature and the trust that existed among members of the TSMSM community. The perceived trust among peers would enable them to reach each other better, as compared to the research team trying to reach TSMSM directly.

“It (RDS) would work best and I think it is one of the recommended methods especially MSM being hidden… of course they will tend to trust information coming from their peers and this group of MSM is actually very highly networked”Jay, student.

“I think it (RDS) would work very well because it’s easier for me to refer someone than you going out there to look for a guy to participate… they (my friends) would believe my word of mouth so it’s convincing enough.”Trevor, student.

Service providers noted that they were already known and trusted by TSMSM, and this existing relationship would make TSMSM referred by the service providers more likely to agree to participate in the prospective survey.

“If you are given something by a friend of yours, it is easy to accept it… but if you are given by a stranger, that is when challenges come in… because you (researchers) are already embedding yourselves in the existing networks, then it is okay because of the already existing trust between us (service providers) and the students.”Justin, service provider.

The proposed double reimbursement for first participating and then getting a friend to participate in the prospective survey also made RDS popular. The first reimbursement would be used to cater for expenses, such as fare to get to the survey site, and the second would motivate peers to bring in their friends to take part in the prospective survey.

“It (RDS) will be good because of the reimbursement… you are using your transport from where you live to the place where you are going to get the service or the research so it will be a good thing. Once you’ve told someone that the moment you bring someone there is something for you it’s not that hard (for a friend to agree to participate).”Danny, student.

“I think since there is a small reimbursement, I think that would motivate people to really recruit their friends to come for the study.”Kyle, student.

### 3.4. Anticipated RDS Implementation Challenges and Proposed Mitigation Strategies

Participants anticipated that RDS implementation would face some challenges and offered probable ways of mitigating these challenges. Key among the anticipated challenges was that if the survey was associated with CBOs that serve sex workers, then TSMSM would not be willing to take part in the survey because they would not want to be associated with sex work. It was proposed that the researchers present themselves to possible participants as independent from these CBOs so as to encourage participation.

“The only challenge will be if this (the survey) is associated with some organizations it might discourage some people from participating. You find this group (TSMSM) sometimes doesn’t want to be associated with some community clinics because they have a reputation of serving sex workers… I guess I like that you presented yourself as an independent researcher from an academic institution, that is straightforward… people will feel free to participate.”Jay, student.

Another foreseen challenge was the possible loss of recruitment coupons. Participants suggested various ways of handling this, including indicating the incentives on the coupons so that TSMSM would be motivated to keep the coupons safe, using coupons of small size that would easily fit in a pocket and placing coupons in an envelope.

“They (coupons) get lost a lot, someone just presents himself without it. If they are communicating the incentive directly, then it works… someone will be able to store it properly.”Jay, student.

“The challenge I was thinking of is when it (the coupon) gets lost… but if it will be in a smaller size, a pocketable size it is not easy to lose it.”Frank, student.

“The person can lose the coupon. If it can be sealed in an envelope well, then it will be easy for them to handle.”Caleb, student.

It was also noted that some TSMSM would be unwilling to participate if they were not assured about confidentiality with regard to their same-sex behavior. To encourage TSMSM recruitment, it was suggested that during consenting, TSMSM be provided with adequate information on how their privacy and confidentiality would be guaranteed, as well as assurance that it was safe to participate in the survey.

“I believe with detailed information, if one got the information correctly, maybe the willingness (to participate) would be much better.”Kyle, student.

“Most of them (MSM), I think you need to assure them of the confidentiality in the interview. That is the thing that most MSM fear. Maybe they will be mentioned out there, or if they are students like at the university, they will be discriminated.”Pitts, student.

To further encourage recruitment, it was suggested that part of the survey team be MSM because TSMSM would be more comfortable opening up to other MSM.

“Due to the sensitivity of what you’re dealing with, only some of my friends would agree to come for the study… most of them are ok opening up to fellow members (MSM) but when it comes to somebody else they don’t open up. So it is good to have other MSM working with you.”Perry, student.

Participants also predicted that TSMSM would encounter a twofold challenge in getting to the survey site: knowing the location of the survey site and having the means to get to the survey site. To allay this challenge, participants proposed that the survey team should consider giving directions to the survey site by sharing a location pin on smart phones, providing a telephone number TSMSM could call if they needed clarifications about the location of the survey site and reimbursing participants with money to cover transport costs.

“In terms of direction, if someone has a smartphone you can send them the location pin [GPS coordinates] of the study office. It could work… instead of walking around asking from people… and include phone details just for inquiries.”Andy, student.

“You might find someone does not have the transport to get to town. If there could be a way they could be sorted out when they come, they get reimbursed the amount… I think it would work. It would give someone motivation to come and participate.”Andy, student.

From the service providers’ point of view, it was noted that due to the proposed incentives in the prospective survey, there was a likelihood of recruiting MSM who were not TSMSM and/or men who were not MSM. To mitigate this to some extent, it was recommended that the research team could ask to see a student identification card and compare the student’s facial appearance with the image on the identification card.

“When someone wants to get the incentive and does not have friends who are students, you are likely to be brought people who are coached… you are also likely to be brought people who are not MSM who are also coached.”Nicole, service provider.

“To participate, one comes with his school ID… but now, one can borrow an ID… it is still not 100% viable (foolproof) but it tries to limit number of non-students participating.”Nicole, service provider.

Finally, carrying out the prospective survey when tertiary learning institutions were closed due to COVID-19 restrictions or when students were having examinations would interfere with recruitment of participants. It was proposed that the prospective survey be conducted at a time when COVID-19 restrictions would have eased and students were not sitting examinations.

“For a student it depends on whether he is currently in college or at home because of corona (COVID-19)… and also, are you giving these coupons during exams time? Of course no one would come if they have exams. And you know students at university, they only read during exams time and therefore they would not bring back the coupons during that time.”Justin, service provider.

“It is better if you do the study when universities re-open (after COVID-19 restrictions have eased) and also when the students are not having exams.”Mike, service provider.

## 4. Discussion

We assessed whether RDS was an appropriate and acceptable strategy for recruiting young TSMSM in an HIV/STI biobehavioral survey. We also assessed the challenges anticipated with the use of this method and strategies to mitigate these challenges as proposed by the participants. RDS relies on certain assumptions that need to be met for it to be considered an appropriate sampling method for a particular population. These assumptions are that (1) the target population is sufficiently networked to sustain recruitment through the waves and eventually reach the desired sample size, (2) peer relationships are reciprocal (a knows and can recruit b; b knows and can recruit a), (3) recruitment within an individual’s network is random and (4) sampling occurs with replacement so that the set of available recruits is not depleted [30]. Our findings demonstrate that the first two assumptions were met for the TSMSM population. To begin with, the TSMSM population was well networked, with members exhibiting close social ties and having large network sizes. Secondly, members of the TSMSM population formed reciprocal relationships characterized by trust between each other. Based on the fulfillment of these two assumptions, RDS would thus be an appropriate recruitment strategy for TSMSM. Besides the trust that exists between peers, RDS was also acceptable to TSMSM due to the proposed double incentives and the discrete approach the method uses.

Our target population of TSMSM is young and socially marginalized. Other studies have shown that, to varying degrees, RDS is an appropriate and acceptable strategy for recruiting university students [31]; sexual and gender minority youth, including young MSM [32]; and young adult non-medical users of pharmaceutical opioids [33] for research. However, other studies have found that RDS may be an inappropriate strategy for recruiting socially marginalized populations, such as urban young MSM [34] and young illicit drug users [35], mainly due to members of the target populations having small network sizes and insufficient ties within their networks, resulting in the inability to sustain recruitment. Notably, the studies highlighted here used data that were collected after the RDS surveys had been completed. To the best of our knowledge, this is the first study to use formative research to assess the appropriateness and acceptability of using RDS to recruit young TSMSM for HIV/STI research in sub-Saharan Africa. As in other studies [36,37,38], this formative research provided us with useful information on the challenges that could be expected with the use of RDS, as well as probable mitigation strategies in response to these challenges. Being aware of these challenges is important to research teams because the teams are then able to put in place measures to address these challenges prior to implementing RDS surveys.

Our study had some limitations. By the nature of its design, formative research aims to describe the attributes of a community that are relevant to a specific public health issue. It is not designed to test hypotheses. Rather, in this instance, formative research helped inform whether RDS could be considered an appropriate and acceptable strategy for the target population and contributed to the RDS survey planning process by identifying anticipated implementation challenges and proposed mitigation strategies for these challenges. Therefore, we cannot say for sure that based on these findings, an RDS survey among TSMSM would be successful. We would have to conduct the RDS survey itself and then evaluate its outcomes and consider the value of the formative research to the RDS survey. Another limitation of this study is that we interviewed a small, convenient sample of service providers. However, we felt that with this sample, given the dual roles each of the providers had at their respective organizations and the day-to-day interactions they had with MSM, we would be able to obtain sufficient information to meet the objectives of our study. Additionally, the three organizations from which the participants were sampled are the primary providers of MSM-exclusive/friendly services and are highly rated by MSM. Furthermore, the study was conducted at a time when there were government-imposed COVID-19 restrictions, with most staff in these organizations working from home, thus limiting their availability and accessibility to participate in the study. In addition, our study only utilized interviews for data collection. Other methods of qualitative data collection, such as focus group discussions and community mapping, may yield more information for assessing the appropriateness and acceptability of the RDS method, as well as anticipated RDS implementation challenges and possible strategies to mitigate these challenges [22,38]. Future studies should consider larger sample sizes, as well as a combination of qualitative methods so as to yield more data. Finally, the study was conducted in the capital city, which is expected to be more liberal than other smaller cities or rural settings in the country, thus limiting the generalizability of our findings. Nonetheless, these findings demonstrate the appropriateness and acceptability of RDS among TSMSM, in addition to bringing to light the anticipated challenges of using RDS and offering recommendations on how these challenges could be addressed prior to implementing the RDS survey.

## 5. Conclusions

TSMSM had large network sizes and strong ties within the networks, properties considered to be appropriate to sustain recruitment through RDS waves. The RDS method was acceptable to TSMSM due to the trust that existed amongst them, making it easier for them to reach each other as compared to the researchers themselves attempting to reach TSMSM directly. The study also offered valuable lessons on the challenges that would be expected with the use of RDS in the TSMSM population and proposed strategies to mitigate these challenges. Pre-assessing the suitability and agreeability of methods to reach young marginalized populations appears to be a practical way to gain entry into these populations, ensuring that the views and voices of possible future participants are incorporated into the design of prospective studies.

## Figures and Tables

**Table 1 ijerph-19-07331-t001:** Demographic characteristics of TSMSM participants (n = 13).

Variable	n (%)
Age	
18–20	3 (23.1)
21–24	10 (76.9)
Level of education	
Undergraduate	13 (100)
Year 1	3 (23.1)
Year 2	5 (38.5)
Year 3	5 (38.5)
Type of institution	
University	11 (84.6)
* TVET/College	2 (15.4)
Ownership of institution	
Public	9 (69.2)
Private	4 (30.8)
Field of study	
Arts and humanities	3 (23.1)
Business and management	3 (23.1)
Engineering and technology	3 (23.1)
Health sciences	2 (15.4)
Social sciences	2 (15.4)
Residence	
Rented (apartment/flat/hostel outside college/university)	8 (61.5)
At home with family	5 (38.5)
Main source of income	
Part time employment	8 (61.5)
Parents/guardians	5 (38.5)
Network size	
10–24	6 (46.2)
25–40	7 (53.8)
Sexual orientation	
Homosexual or gay	9 (69.2)
Bisexual	4 (30.8)

* TVET: technical and vocational education training institution.

## Data Availability

The data presented in this study are available on request from the corresponding author. The data are not publicly available because they are qualitative interviews that contain personal and potentially identifiable information from members of a marginalized population.

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
