# Peer review of "“My Friends Would Believe My Word”: Appropriateness and Acceptability of Respondent-Driven Sampling in Recruiting Young Tertiary Student Men Who Have Sex with Men for HIV/STI Research in Nairobi, Kenya"

_ijerph, 2022, doi:10.3390/ijerph19127331_

Round 1
Reviewer 1 Report
“My friends would believe my word”: appropriateness and acceptability of respondent-driven sampling in recruiting young tertiary student men who have sex with men for HIV/STI research in Nairobi, Kenya
I don’t believe this is a relevant contribution to the journal’s readers. Notions of appropriateness and acceptability of a sampling method aren’t clear, and the sample is too small, even for a qualitative study.
Best wishes.
Reviewer 2 Report
The study concerns an issue that is extremely difficult to assess, due to the fact that homosexuality is illegal in Kenya, so any research on this subject must be fragmentary. Due to the scientific workshop, the work is correct, although it requires corrections. The reviewer assesses the identification of the research problem as good, but does not find the purpose of the research in the text. The work should clearly indicate what its purpose is, because after reading the work, you may have doubts about it.
The work uses scientific research methodologies and questionnaires that do not raise doubts about their reliability and adequacy. In addition, the authors used a reliable coding system that follows the rules of scientific inference.
The reviewer asks for additional clarification of the issue. If MSM behavior is criminalized in Kenya, how did you get approval from local ethics boards when one of the questions was about criminals ("do you know any homosexual person")? Does the respondent who is asked such a question and does he know that the answer may involve punishment, will he answer honestly?
It must be important for the work to precisely define the goal, that is to say whether the goal is to conduct an HIV exposure study on MSM students or whether the predisposition of this environment is influenced by some research tools on this topic. It is illegible at work.
The study used a very small group of respondents, but the reviewer understands the legal constraints that affect it and is taking this report as a pilot.
The obtained answers should be grouped, try to parameterize. They are presented as free statements which, although related to each other, are not thematically related. It is recommended to parameterize them in order to facilitate the discussion.
A few specific words are missing from the conclusions, as evidenced by the research. What are the conclusions that will be discussed later.
Round 2
Reviewer 1 Report
Justifications do not make the article's overall merit higher. I still believe this contribution isn't relevant to the journals readers. Therefore, I leave it to the Editor to decide wether or not to further go with the process of revision. If so, I would like to be withdrawn from it. Thank you.
Reviewer 2 Report
The article was corrected in accordance with the reviewer's guidelines, although the specification of the purpose turned out to be incorrect. It has been described as two purposes, of which it is not indicated which is the main and which is complementary. In the second goal "identification of potential challenges that we are likely to encounter using the RDS 110 method", the authors set another, third goal: to identify possible solutions to these challenges. Such a goal cannot be scientifically achieved. In the cover letter, the responses seem credible to the reviewer's comments and I accept them.Author Response
Please see the attachment
